# CALE: Continuous Arcade Learning Environment

**Jesse Farebrother**
McGill University
Mila - Québec AI Institute
Google DeepMind
jfarebro@cs.mcgill.ca

**Pablo Samuel Castro**
Google DeepMind
Université de Montréal
Mila - Québec AI Institute
psc@google.com

## Abstract

We introduce the Continuous Arcade Learning Environment (CALE), an extension of the well-known Arcade Learning Environment (ALE) [Bellemare et al., 2013]. The CALE uses the same underlying emulator of the Atari 2600 gaming system (Stella), but adds support for continuous actions. This enables the benchmarking and evaluation of continuous-control agents (such as PPO [Schulman et al., 2017] and SAC [Haarnoja et al., 2018]) and value-based agents (such as DQN [Mnih et al., 2015] and Rainbow [Hessel et al., 2018]) on the same environment suite. We provide a series of open questions and research directions that CALE enables, as well as initial baseline results using Soft Actor-Critic. CALE is available as part of the ALE at https://github.com/Farama-Foundation/Arcade-Learning-Environment.

## 1 Introduction

Generally capable autonomous agents have been a principal objective of machine learning research, and in particular reinforcement learning, for many decades. *General* in the sense that they can handle a variety of challenges; *capable* in that they are able to "solve" or perform well on these challenges; and they are able to learn *autonomously* by interacting with the system or problem by exercising their *agency* (e.g. making their own decisions). While deploying and testing on real systems is the ultimate goal, researchers usually rely on academic benchmarks to showcase their proposed methods. It is thus crucial for academic benchmarks to be able to test generality, capability, and autonomy.

Bellemare et al. [2013] introduced the Arcade Learning Environment (ALE) as one such benchmark. The ALE is a collection of challenging and diverse Atari 2600 games where agents learn by directly playing the games; as input, agents receive a high dimensional observation (the "pixels" on the screen), and as output they select from one of 18 possible actions (see Section 2). While some research had already been conducted on a few isolated Atari 2600 games [Cobo et al., 2011, Hausknecht et al., 2012, Bellemare et al., 2012], the ALE's significance was to provide a unified platform for research and evaluation across more than 100 games. Using the ALE, Mnih et al. [2015] demonstrated, for the first time, that reinforcement learning (RL) combined with deep neural networks could play challenging Atari 2600 games with super-human performance. Much like how ImageNet [Deng et al., 2009] ushered in the era of Deep Learning [LeCun et al., 2015], the Arcade Learning Environment spawned the advent of Deep Reinforcement Learning.

In addition to becoming one of the most popular benchmarks for evaluating RL agents, the ALE has also evolved with new extensions, including stochastic transitions [Machado et al., 2018], various game modes and difficulties [Machado et al., 2018, Farebrother et al., 2018], and multi-player support [Terry and Black, 2020]. What has remained constant is the suitability of this benchmark for testing *generality* (there is a wide diversity of games), *capability* (many games still prove challenging for most modern agents), and *agency* (learning typically occurs via playing the game).

38th Conference on Neural Information Processing Systems (NeurIPS 2024) Track on Datasets and Benchmarks.

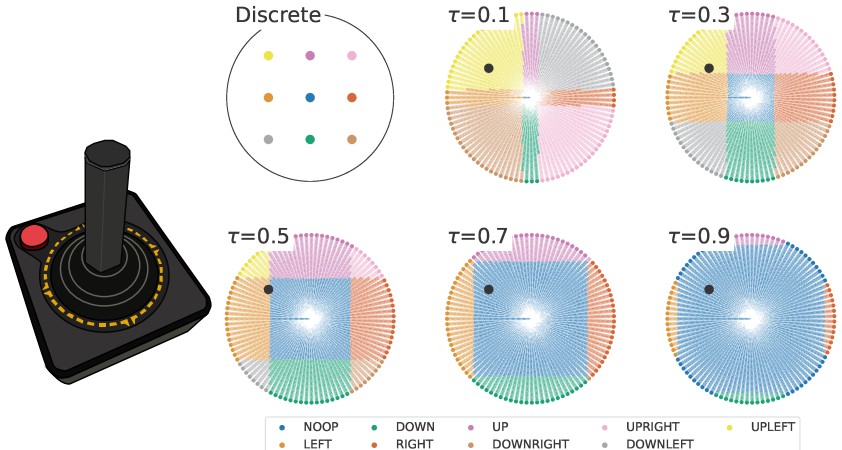

Figure 1: **Left panel:** Atari CX10 controller. **Right panel:** Discrete joystick positions (top left) versus continuous joystick positions with varying values of the threshold $\tau$. The black circle corresponds to a joystick at position $(r, \theta) = (0.61, 2.53)$.

There are a number of design choices that have become standard when evaluating agents on the ALE, and which affect the overall learning dynamics. These choices involve modifying the temporal dynamics through frame skipping; adjusting the input observations with frame stacking, grey-scaling, and down-sampling; and converting the range of joystick movements into a standardized set of 18 discrete actions to be shared across all games.[1] The design of the action space resulted in a rather profound impact on the type of research conducted on the ALE. In particular, *it is only compatible with discrete-action agents*. This has led to certain classes of agents, often based on Q-learning [Watkins, 1989], to focus primarily on the ALE. On the other hand, agents based on policy gradient [Sutton et al., 1999] or actor-critic [Konda and Tsitsiklis, 1999] methods, while sometimes evaluating on the ALE by using discrete variants, tend to focus on entirely different benchmarks, such as MuJoCo [Todorov et al., 2012] or DM-Control [Tassa et al., 2018].

In this paper, we introduce the Continuous Arcade Learning Environment (CALE) that introduces a continuous action space making for an interface that more closely resembles how humans interact with the Atari 2600 console. Our work enables the evaluation of both discrete and continuous action agents on a single unified benchmark, providing a unique opportunity to gain an understanding of the challenges associated with the action-space of the agent. Additionally, we present baselines with the popular Soft-Actor Critic [SAC; Haarnoja et al., 2018] algorithm that underscore the need for further research towards general agents capable of handling diverse domains. Finally, we identify key challenges in representation learning, exploration, transfer, and offline RL, paving the way for more comprehensive research and advancements in these areas.

## 2   From Atari VCS to the Arcade Learning Environment

The Atari Video Computer System (VCS), later renamed the Atari 2600, is a pioneering gaming console developed in the late 1970s that aimed to bring the arcade experience to the home. Game designers had to operate under a variety of constraints, including writing code that could execute in time with the electron beam displaying graphics on the CRT screen and rendering graphics using the limited set of primitives provided by the system. Although designed to support a variety of controllers, the majority of games were played with an Atari CX10 "digital" controller (see left panel of Figure 1). Players move a joystick along two axes to trigger one of nine discrete events (corresponding to three positions on each axis) on the Atari VCS. Combined with a "fire" button, this results in 18 possible events the user could trigger.[2]

---

[1]Certain games, such as Pong and Breakout, were originally played using a different set of paddle controllers, but were given the same action space in the ALE.

[2]For the interested reader, Montfort and Bogost [2009] provide a great historical overview of the design and development of the Atari VCS.

The Atari 2600 was one of the first widely popular home gaming devices and even became synonymous with "video games", marking the beginning of exponential growth in the video game industry over the following decades. A likely reason for its popularity was the use of external cartridges containing read-only memory (ROM), which allowed for a scalable plug and play experience. Over 500 games were developed for the original console, offering a wide variety of game dynamics and challenges that appealed to an ever-growing audience. As personal computing became more widespread, emulators such as Stella [Mott et al., 1996] emerged, allowing enthusiasts to continue playing Atari 2600 games without needing the original hardware.

Building upon the Stella emulator, Bellemare et al. [2013] introduced the Arcade Learning Environment (ALE) as a challenging and diverse environment suite for evaluating generally capable agents. The authors argue the ALE contains three crucial features which render it a meaningful baseline for agent evaluation: **variety** – it contains a diverse set of games; **relevance** – the varied challenges presented are reflective of challenges agents may face in practically-relevant environments; and **independence** – it was developed independently for human enjoyment, free from researcher bias.

This seminal benchmark was used by Mnih et al. [2015] to showcase super-human performance when combining temporal-difference learning [Sutton, 1984] with deep neural networks. The performance of their DQN agent was compared against the average performance of a single human expert; these average human scores now serve as the standard way to normalize and aggregate scores on the ALE [Agarwal et al., 2021]. Since its introduction, numerous works have improved on DQN, such as Double DQN [Hasselt et al., 2016], Rainbow [Hessel et al., 2018], C51 [Bellemare et al., 2017], A3C [Mnih et al., 2016], IMPALA [Espeholt et al., 2018], R2D2 [Kapturowski et al., 2019], and Agent57 [Badia et al., 2020]; the ALE continues to serve as a simulator-based test-bed for evaluating new algorithms and conducting empirical analyses, especially with limited compute budgets.

## 3 CALE: Continuous Arcade Learning Environment

The original Atari CX10 controller (left panel of Figure 1) used a series of pins to signal to the processor when the joystick is in one of nine distinct positions, visualized in the 'Discrete' sub-panel in Figure 1 [Sivakumaran, 1986]. When combined with a boolean "fire" button, this results in 18 distinct joystick *events*. Indeed, player control in the Stella emulator is built on precisely these distinct events [Mott et al., 1996], and they also correspond to the 18 actions chosen by the ALE.

However, although the resulting events are discrete, the range of joystick motion available to players is continuous. We add this capability by introducing the **C**ontinuous **A**rcade **L**earning **E**nvironment (CALE), which switches from a set of 18 discrete actions to a three-dimensional continuous action space. Specifically, we use the first two dimensions to specify the polar coordinates $(r, \theta)$ in the unit circle corresponding to all possible joystick positions, while the last dimension is used to simulate pressing the "fire" button. Concretely, the action space is $[0, 1] \times [-\pi, \pi] \times [0, 1]$. The implementation of CALE is available as part of the ALE at https://github.com/Farama-Foundation/Arcade-Learning-Environment (under GPL-2.0 license). See Appendix A for usage instructions.

As in the original CX10 controller, this continuous action space still needs to trigger discrete events. For this, we use a threshold $\tau$ to demarcate the nine possible position events the joystick can trigger. Figure 1 illustrates these for varying values of $\tau$, as well as the different events triggered when the joystick is at position $(r, \theta) = (0.61, 2.53)$. As can be seen, lower values of $\tau$ result in more sensitive control, while higher values can result in less responsive controllers, even to the point of completely occluding certain events (the corner events are unavailable when $\tau = 0.9$, for example).

It is worth highlighting that, since CALE is essentially a wrapper around the original ALE, it is *only* changing the agent action space. Since both discrete and continuous actions ultimately trigger the same events, the underlying game mechanics and learning environment remain unchanged. This is an important point, as it means that we now have a *unified* benchmark on which to directly compare discrete and continuous control agents.

An important difference is that the ALE supports "minimal action sets", which reduce the set of available actions from 18 to the minimum required to play the game. For example, in Breakout only the LEFT, RIGHT, and FIRE events have an effect on game play, resulting in a minimal set of 4 actions. By default, minimum action sets are enabled in the ALE and used by many existing

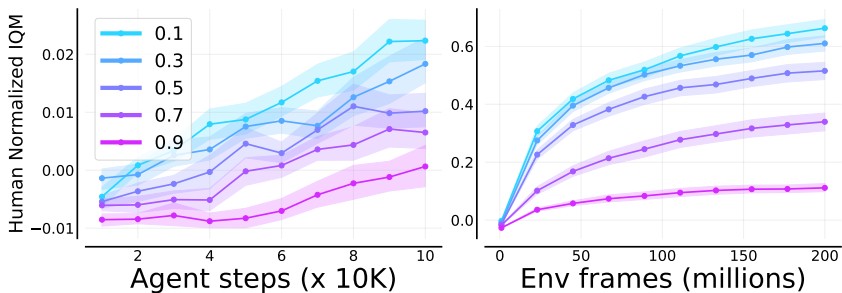

Figure 2: CALE comparison with varying $\tau$ on the 100k (left) and 200m (right) training regimes.

implementations [Castro et al., 2018]. Given the manner in which continuous actions have been parameterized, this minimal action set is unavailable when running with the CALE. Thus, for many games, continuous-action agents trained on the CALE may be at a disadvantage when compared with discrete-action agents trained on the ALE (see comparison in Section 4.5 and Figure 7 in particular). For completeness, we list the minimum action sets for all 60 games in Appendix D.

## 4 Baseline results

We present a series of baseline results on CALE using the soft actor-critic agent [SAC; Haarnoja et al., 2018]. SAC is an off-policy continuous-control method that modifies the standard Bellman backup with entropy maximization [Ziebart et al., 2008, Ziebart, 2010]. DQN and the agents derived from it are also off-policy methods, thereby rendering SAC a more natural choice for this initial set of baselines than other continuous control methods such as PPO. We use the SAC implementation and experimental framework provided by Dopamine [Castro et al., 2018]. We detail our experimental setup and hyper-parameter selection below, and provide further details in Appendix C.

### 4.1 Experimental setup

We use the evaluation protocol proposed by Machado et al. [2018]. Namely, agents are trained for 200 million frames with "sticky actions" enabled, 4 stacked frames, a frame-skip of 4, and on 60 games. Additionally, we use the Atari 100k benchmark introduced by Łukasz Kaiser et al. [2020], which evaluates agents using only 100,000 agent interactions (corresponding to 400,000 environment steps due to frame-skips) over 26 games. The Atari 100k benchmark has become a popular choice for evaluating the sample efficiency of RL agents [D'Oro et al., 2023, Schwarzer et al., 2023]. We follow the evaluation protocols of Agarwal et al. [2021] and report aggregate results using interquartile mean (IQM), with shaded areas representing 95% stratified bootstrap confidence intervals. All experiments were run on P100 GPUs; the 200M experiments took between 5-7 days to complete training, while the 100K experiments took between 1 and 2 hours to complete.

### 4.2 Threshold selection

As mentioned in Section 3, the choice of threshold $\tau$ affects the overall performance of the agents. Consistent with intuition, Figure 2 demonstrates that higher values of $\tau$ result in degraded performance. For the remaining experimental evaluations we set $\tau$ to 0.5. This choice has consequences for SAC, due to the way its action outputs are initialized, which we discuss in the next subsection.

### 4.3 Network architectures

Given an input state $x \in \mathcal{X}$, the neural networks used by actor-critic methods usually consist of an "encoder" $\phi : \mathcal{X} \to \mathbb{R}^d$, and actor and critic heads $\psi_A : \mathbb{R}^d \to \mathcal{A}$ and $\psi_C : \mathbb{R}^d \to \mathbb{R}$, respectively, where $\mathcal{A}$ is the (continuous) action space. Typically the action outputs are assumed to be Gaussian distributions with mean $\mu$ and standard deviation $\sigma$. Thus, for a state $x$, the *value* of the state is $\psi_C(\phi(x))$ and the action selected is distributed according to $\psi_A(\phi(x))$.

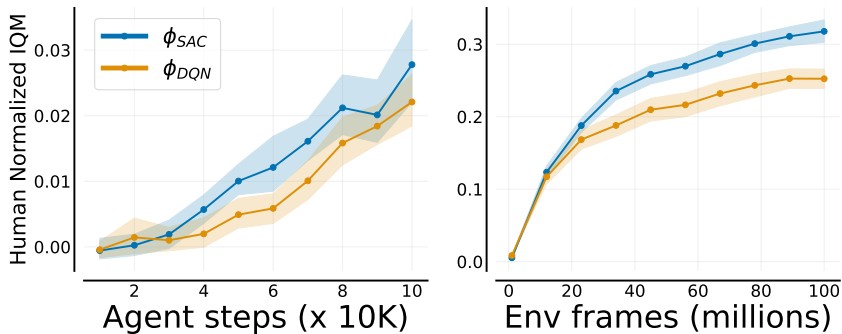

Figure 3: CALE comparison of $\phi_{SAC}$ and $\phi_{DQN}$ on the 100k (left) and 200m (right) training regimes.

The SAC implementation we use initializes $\mu$ in the middle of the action ranges. Thus, for the CALE action space, $\mu$ is initialized at $(0.5, 0.0, 0.5)$. With $\tau = 0.5$, this means the $r$ and "fire" dimensions will be initially straddling the threshold, where action variations are most significant. On the other hand, this initialization produces an initial $\theta$ value of $0.0$, which results in an initial bias towards the RIGHT event (since polar coordinates $(0.5, 0.0)$ correspond to $(0.5, 0.0)$ Cartesian coordinates). See Figure 7 and the surrounding discussion for more details.

For all our experiments we use a two-layer multi-layer perceptron (MLP) with 256 hidden units each for both $\psi_A$ and $\psi_C$. Haarnoja et al. [2018] benchmarked SAC on non-pixel environments, where $\phi$ consisted purely of an MLP. For pixel-based environments like the ALE, however, convolutional networks are typically preferred encoders. Yarats et al. [2021b] proposed a convolutional encoder network for SAC (based on the encoder proposed by Tassa et al. [2018] for the DeepMind Control Suite), which was further used by Yarats et al. [2021a]. We refer to this encoder as $\phi_{SAC}$. We refer to the three-layer convolutional encoder originally used by DQN [Mnih et al., 2015] (and used by most DQN-based algorithms) as $\phi_{DQN}$.

As Figure 3 demonstrates, $\phi_{SAC}$ outperforms $\phi_{DQN}$ in both the 100K and 200M training regimes. Although DQN has not been explicitly tested with $\phi_{SAC}$, it begs the question of whether certain algorithms benefit from certain types of encoder architectures over others; this relates to questions of representation learning, which we discuss below.

## 4.4 Exploration strategies

Due to its objective including entropy maximization and the fact that the actor is parameterized as a Gaussian distribution, SAC induces a natural exploration strategy obtained by sampling from $\psi_A$ (and simply using $\mu$ when acting greedily). We refer to this as the **standard** exploration strategy. However, the exploration strategy typically used on the ALE is $\epsilon$-**greedy**, where actions are chosen randomly with probability $\epsilon$; a common choice for ALE experiments is to start $\epsilon$ at $1.0$ and decay it to $0.01$ over the first million environment frames. For our continuous action setup we sample uniformly randomly in $[0, 1] \times [-\pi, \pi] \times [0, 1]$ with probability $\epsilon$. Perhaps surprisingly, **standard** outperforms $\epsilon$-**greedy** exploration in the 200 million training regime, as demonstrated in Figure 4. This may be due to the way the action outputs are parameterized, and merits further inquiry.

## 4.5 Comparison to existing discrete-action agents

We compare the performance of our SAC baseline against DQN in the 200 million training regime, given that both are off-policy methods which have similar value estimation methods; for the 100k training regime we compare against Data-Efficient Rainbow [DER; Van Hasselt et al., 2019], a popular off-policy method for this regime that is based on DQN. As Figure 6 shows, SAC dramatically underperforms, relative to both these methods. While there may be a number of reasons for this, the most likely one is the fact that SAC was not tuned for CALE, whereas both DER and DQN were tuned specifically for the ALE.

We additionally compared to a version of SAC with a categorical action parameterization which allows us to run it on the original ALE. The hyper-parameters (listed in Appendix C) are based on

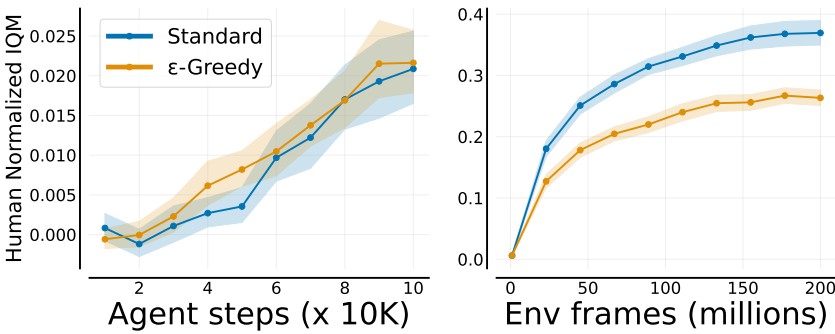

Figure 4: CALE comparison of default SAC exploration with the more common $\epsilon$-greedy exploration used in discrete action agents on the 100k (left) and 200m (right) training regimes.

those suggested by Christodoulou [2019]. Surprisingly, this discrete-action SAC on the ALE agent dramatically underperforms even against our continuous-action SAC on the CALE.

Aggregate performance curves can often conceal interesting per-game differences. Indeed, Figure 5 demonstrates that SAC can sometimes surpass the performance of DQN (Asteroids, Bowling, Centipede), sometimes have comparable performance (Asterix, Boxing, MsPacman, Pong), and sometimes under-perform (BankHeist, Breakout, SpaceInvaders). Minimal action sets (as discussed in Section 3) do not appear to correlate with these performance differences (Bowling, Pong and SpaceInvaders all use a minimal set of 6 actions in the ALE); similarly, reward distributions (as we will discuss below) do not appear to correlate with performance differences between these two agents either. The differences may be due to differences in transition dynamics, as well as exploration challenges, which we discuss below.

Figure 7 displays the distribution of discrete joystick events triggered by both DER and SAC and confirms that, while some games like Breakout on the ALE only trigger 4 events, most events are triggered on the CALE. It is interesting to observe that, as discussed in Section 4.3, SAC has a bias towards the RIGHT action, due to the action parameterization and initialization.

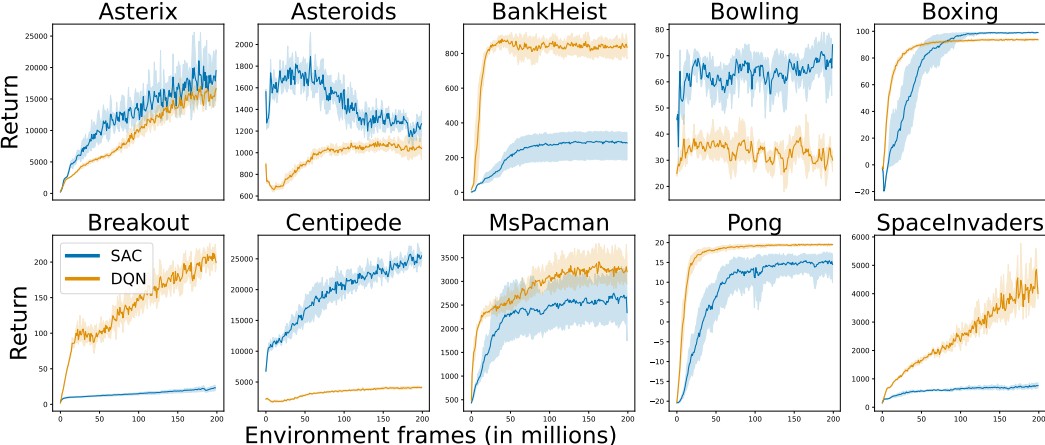

Figure 5: CALE comparison of SAC with DQN (using the default Dopamine implementation [Castro et al., 2018]) on a selection of games. Returns averaged over 5 independent runs, with shaded areas representing 95% confidence intervals.

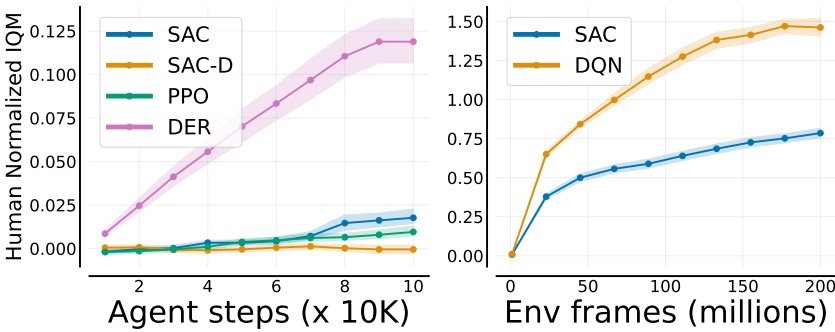

Figure 6: Aggregate comparison of SAC and PPO on the CALE with DER and SAC-D on the ALE [Van Hasselt et al., 2019] (left), and DQN on the the ALE [Mnih et al., 2015] (right).

## 5 Comparison to other continuous control environments

The most commonly used continuous control methods are centered around robotics tasks such as locomotion [Todorov et al., 2012, Wołczyk et al., 2021, Khazatsky et al., 2024], where transition dynamics are relatively smooth and can thus be approximated reasonably well with Gaussian distributions. This assumption is often critical to certain methods, for instance in the reparameterization of the Wasserstein-2 for the DBC algorithm proposed by Zhang et al. [2021]). Thus, the "non-smoothness" of the CALE yields a novel challenge for continuous control agents, which could help us better understand, and improve, them.

Additionally, the reward structures in these environments tend to be much denser than in the ALE. In Figure 8 we plot the reward distributions for an exploratory policy in both the Arcade Learning Environment [Bellemare et al., 2013] and the DeepMind Control Suite [DMC; Tunyasuvunakool et al., 2020]. Specifically, for the ALE we take the rewards collected in the first 1M frames for all games in the RL Unplugged dataset [Gulcehre et al., 2020] corresponding to the exploratory phase of a DQN agent. For DMC we leverage the ExoRL dataset [Yarats et al., 2022] and collect rewards on the Cheetah, Walker, Quadruped, and Cartpole domains from an exploratory random network distillation policy. Figure 8 shows that the proportion of rewards that are 0 in Atari is higher than in most of the DMC tasks, indicating that rewards are relatively more sparse in Atari.

In addition to robotics/locomotion tasks, there have been a number of recent environments simulating real-world continuous control scenarios. These include optimal control problems (continuous in both time and space) [Howe et al., 2022, Ma et al., 2024], simulated industrial manufacturing and process control [Zhang et al., 2022], power consumption optimization [Moriyama et al., 2018], process control [Bloor et al., 2024], dynamic algorithm configuration [Eimer et al., 2021], among others.

## 6 Research directions

Since its release, the Arcade Learning Environment has been extensively used by the research community to explore fundamental problems in decision making. However, most of this research has focused specifically on value-based methods with discrete action spaces. On the other hand, many of the challenges presented by the ALE, such as exploration and representation learning, are not always central to existing continuous control benchmarks (see discussion in Section 5). In this section, we identify several research directions that the CALE facilitates. While many of these questions can be explored in different environments, the advantage of the CALE is that it has a *direct* analogue in the ALE, thereby enabling a more direct comparison of continuous- and discrete-control methods.

**Exploration** As discussed in Section 4.4, $\epsilon$-greedy is the default exploration strategy used by discrete-action agents on the ALE. Despite the existence of a number of more sophisticated methods, Taiga et al. [2020] argues that these were over-fit to well-known hard exploration games such as Montezuma's Revenge; they demonstrated that, when aggregating with easier exploration games, $\epsilon$-greedy out-performs the more sophisticated methods. In contrast, the results in Section 4.4 demonstrate that $\epsilon$-greedy under-performs simply sampling from $\mu$ in SAC. This may be an instance

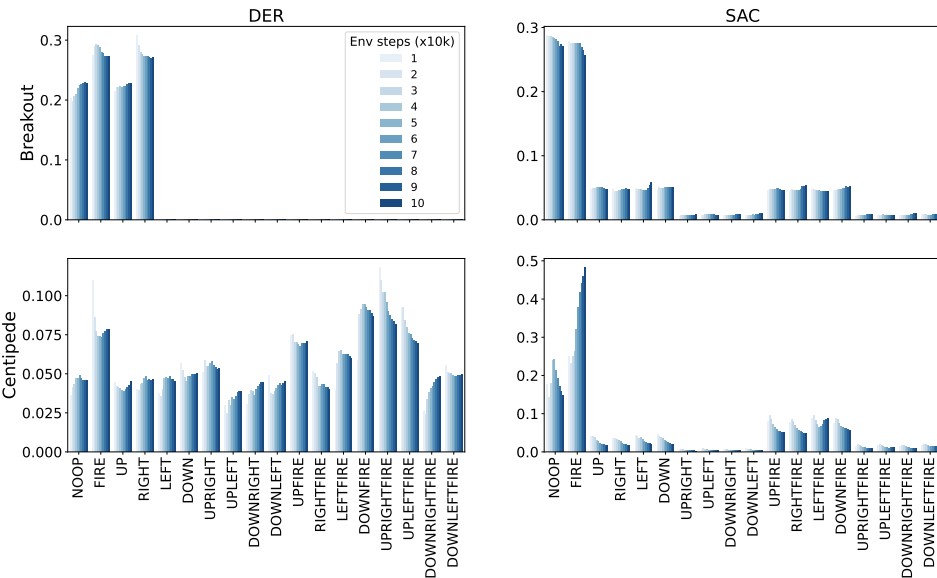

Figure 7: Comparison of joystick event distributions during training of DER on the ALE (left column) and SAC on the CALE (right column) in the 100K benchmark. These are on a single run when training on Breakout (where DQN strongly outperforms SAC) and Centipede (where SAC strongly outperforms DQN).

of *policy churn*, which has been shown to have implicit exploratory benefits in discrete-action agents [Schaul et al., 2022]. Interestingly, our results show that for SAC-D (SAC with discrete actions explored in Section 4.5), $\epsilon$-greedy outperforms sampling from the categorical distribution for exploration (see Appendix F). We believe the CALE provides a novel opportunity for developing exploration methods for continuous-control agents in non-robotics tasks.

**Network architectures**    Recent work has demonstrated the value in exploring alternative network architectures for RL agents [Espeholt et al., 2018, Graesser et al., 2022, Obando-Ceron et al., 2024a,b]. Similarly, notions of "learned representations" [Schwarzer et al., 2020, Castro et al., 2021, Zhang et al., 2021, Yarats et al., 2021b, Farebrother et al., 2023] may benefit from different techniques based on the type of action space and losses used (a fact confirmed by the results in Figure 3). Indeed, Farebrother et al. [2024] demonstrated a stark performance difference resulting from switching from regression to classification losses; given their evaluations was limited to value-based discrete-action agents, it remains an open question whether similar findings carry over to continuous action spaces.

**Offline RL**    Offline RL, where RL agents trained on a fixed dataset [Levine et al., 2020], has seen a significant growth in interest over the last few years. One of the main challenges in this setting is when there is insufficient state-action coverage in the dataset; this is particularly pronounced in discrete-action settings, where there is no clear notion of *similarity* between actions. Continuous control settings perhaps do provide a more immediate notion of action similarity, which could help mitigate out-of-distribution issues in offline RL. For instance, would the tandem effect [Ostrovski et al., 2021] still be present when training from offline data in continuous action spaces?

**Plasticity**    Nikishin et al. [2022] demonstrated that SAC benefits from full network resets in MuJoCo, where a multi-layer perceptron network is used. For SPR [Schwarzer et al., 2020] on the 100k ALE, the authors originally had to limit resets to the penultimate (dense) layer; only by switching to shrink and perturb [Ash and Adams, 2020] does this network benefit from "resets" [D'Oro et al., 2023, Schwarzer et al., 2023]. An interesting question is whether the benefit of full resets are tied to the use of a continuous-control actor-critic method like SAC, or to the fact that only dense layers are needed for MuJoCo. More generally, do findings related to plasticity loss [Sokar et al., 2023, Lyle et al., 2023] apply equally to discrete- and continuous-control agents?

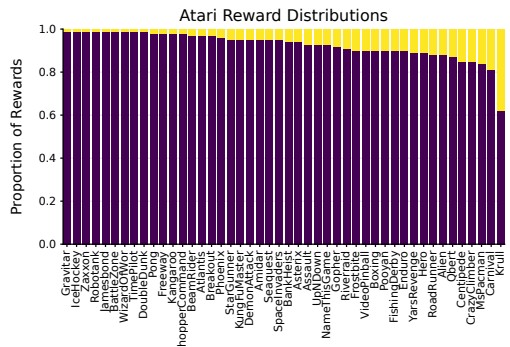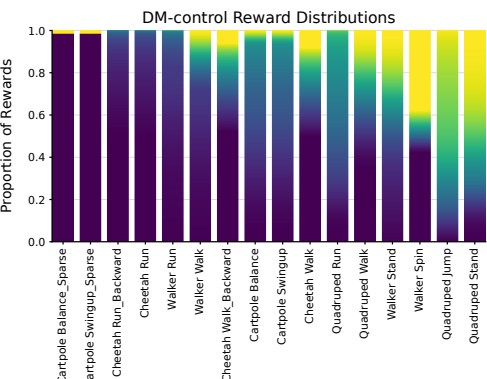

Figure 8: Comparison of reward distributions between ALE (left) and the DM-control (right). For each environment 1M rewards are collected from exploratory agents. Each color in the plot corresponds to a reward value between 0 and 1 with the height of that color corresponding to the relative proportion of that reward in the dataset, i.e., the quantile function of the empirical reward distribution.

**Action parameterizations** The choice of Gaussian distributions for each of the action dimensions, initialized in the middle of the action ranges (as used by SAC) is by no means the only option. It would be interesting to explore alternative action parameterizations, different inductive biases, and evaluate agents already making use of similar re-parameterizations [Hafner et al., 2020].

# 7 Discussion

Academic benchmarks in machine learning are meant to provide a standardized and reproducible methodology with which to evaluate and compare algorithmic advances. In RL, these benchmarks have historically been divided between those suitable for discrete control (such as the ALE), and those suitable for continuous control (such as MuJoCo and DM-Control)[3]. This has made it difficult to directly compare the performance of these two types of algorithms, resulting in less transfer of advances between the continuous- and discrete-control communities than one would hope for.

One of the advantages of the CALE is that it provides a *unified* suite of environments on which to evaluate both types of algorithms, given that both the ALE and the CALE use the same underlying joystick events and Stella emulator. The ALE has been used in a large number of research papers, and there is a growing sentiment that it is no longer interesting; the CALE provides a fresh take on this benchmark, while benefiting from the familiarity that the community already has with it.

One could argue that human evaluations, introduced by Mnih et al. [2015] and used to normalize most ALE experiment scores, are more relevant with the CALE since the human evaluator presumably played on a real joystick. Given that our SAC baseline achieves only 0.4 IQM (where a 1.0 indicates human-level performance), the CALE provides a new challenge to achieve human-level performance on the suite of Atari 2600 games, and aid in the development of generally capable agents.

**Limitations** One limitation of this work is the number of baselines evaluated. We used the Dopamine framework [Castro et al., 2018] for our evaluations, which unfortunately only provides SAC and a recently added implementation of PPO as continuous-control agents. It would be useful to evaluate other continuous-control agents, as well as other agent implementations, on the CALE to build a broader set of baselines for future research. While most games use the joystick illustrated in Figure 1, Pong and Breakout were originally played on non-discrete *paddles* [Montfort and Bogost, 2009]; for this version of the CALE we decided to maintain the same action dynamics across all games, but it would be interesting to add support for paddles, where continuous actions are no longer mapped to discrete events.[4]

---

[3]Tang and Agrawal [2020] showed that discretizing actions can improve performance on DMC tasks.

[4]In the ALE, discrete actions are mapped to hard-coded paddle displacements, which we replicated in our implementation.

**Acknowledgements** The authors would like to thank Georg Ostrovski for providing us with a valuable review of an initial version of this work. Additionally, we thank Hugo Larochelle, Marc G. Bellemare, Harley Wiltzer, Doina Precup, and the Google DeepMind Montreal team for helpful discussions during the preparation of this submission. We would also like to thank the Python community [Van Rossum and Drake Jr, 1995, Oliphant, 2007] for developing tools that enabled this work, including NumPy [Harris et al., 2020], Matplotlib [Hunter, 2007] and JAX [Bradbury et al., 2018]. Finally, we would like to thank Mark Towers and Jet and the Farama Foundation for their help reviewing the code to integrate CALE into the ALE.

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

## A  How to run CALE

CALE is included as of version 0.10 of the Arcade Learning Environment [Bellemare et al., 2013] which can be installed with the command `pip install ale-py`. A Gymnasium [Towers et al., 2023] interface is also provided and can be installed via `pip install gymnasium[atari]`. Once installed the keyword argument `continuous` can enable continuous actions as shown in Listing 1.

```python
import gymnasium

# `env.action_space` will be continuous
env = gymnasium.make("Pong-v5", continuous=True)
```

Listing 1: Enabling continuous action spaces in the Arcade Learning Environment [Bellemare et al., 2013] via the Gymnasium [Towers et al., 2023] Python interface.

## B  Code specifications

The implementation of CALE is available as part of the ALE: https://github.com/Farama-Foundation/Arcade-Learning-Environment (under GPL-2.0 license).

For SAC and PPO, we used the Dopamine [Castro et al., 2018] implementations. Taking Dopamine's root directory https://github.com/google/dopamine/, the specific code paths used are:

- The SAC implementation is available at labs/cale/sac_cale.py
- The PPO implementation is available at labs/cale/ppo_cale.py
- All networks used are available at labs/cale/networks.py
- For SAC-D we simply modified the SAC actor outputs to emit a categorical distribution with `jax.random.categorical`. From this, we can easily extract the log probabilities with `jax.nn.log_softmax`, and select actions greedily with `jnp.argmax`.

## C  Hyper-parameters

In the following table we specify the hyper-parameters used for the various agents considered. For the most part we used the default hyper-parameters specified in the Dopamine gin files for DER, DQN, and SAC. For SAC-D, we modified settings according to what was suggested by Christodoulou [2019].

The full hyper-parameter specifications for SAC are available at labs/cale/configs/sac_cale.gin and labs/cale/configs/sac_cale_100k.gin.

The full hyper-parameter specifications for PPO are available at labs/cale/configs/ppo_cale.gin and labs/cale/configs/ppo_cale_100k.gin.

Table 1: Hyper-parameter setting for all agents.

| Hyper-parameter | DER | DQN | SAC | SAC-D | PPO |
|---|---|---|---|---|---|
| Adam $\epsilon$ | 0.00015 | 1.5e-4 | 1.5e-4 | 1.5e-4 | 1e-5 |
| Batch Size | 32 | 32 | 32 | 64 | 1024 |
| Number of hidden units | 512 | 512 | 512 | 512 | 512 |
| Discount Factor | 0.99 | 0.99 | 0.99 | 0.99 | 0.99 |
| Learning Rate | 0.0001 | 6.25e-5 | 6.25e-5 | 0.0003 | 2.5e-4 |
| Exploration $\epsilon$ | 0.01 | 0.01 | 0.01 | 0.01 | 0.01 |
| Minimum Replay History | 1600 | 20000 | 20000 | 20000 | - |
| Update Horizon | 10 | 1 | 1 | 1 | - |
| Update Period | 1 | 4 | 4 | 4 | - |

## D  ALE game specifications

In the following list we indicate the minimum action values for each game. Games with an asterisk next to them are games which are part of the 26 games for the Atari 100K benchmark [Łukasz Kaiser et al., 2020].

- AirRaid (6)
- Alien* (18)
- Amidar* (10)
- Assault* (7)
- Asterix* (9)
- Asteroids (14)
- Atlantis (4)
- BankHeist* (18)
- BattleZone* (18)
- BeamRider (9)
- Berzerk (18)
- Bowling (6)
- Boxing* (18)
- Breakout* (4)
- Carnival (6)
- Centipede (18)
- ChopperCommand* (18)
- CrazyClimber* (9)
- DemonAttack* (6)
- DoubleDunk (18)
- ElevatorAction (18)
- Enduro (9)
- FishingDerby (18)
- Freeway* (3)
- Frostbite* (18)
- Gopher* (8)
- Gravitar (18)
- Hero* (18)
- IceHockey (18)
- Jamesbond* (18)
- JourneyEscape (16)
- Kangaroo* (18)
- Krull* (18)
- KungFuMaster* (14)
- MontezumaRevenge (18)
- MsPacman* (9)
- NameThisGame (6)
- Phoenix (8)
- Pitfall (18)

- Pong* (6)
- Pooyan (6)
- PrivateEye* (18)
- Qbert* (6)
- Riverraid (18)
- RoadRunner* (18)
- Robotank (18)
- Seaquest* (18)
- Skiing (3)
- Solaris (18)
- SpaceInvaders (6)
- StarGunner (18)
- Tennis (18)
- TimePilot (10)
- Tutankham (8)
- UpNDown* (6)
- Venture (18)
- VideoPinball (9)
- WizardOfWor (10)
- YarsRevenge (18)
- Zaxxon (18)

# E    Per-game results

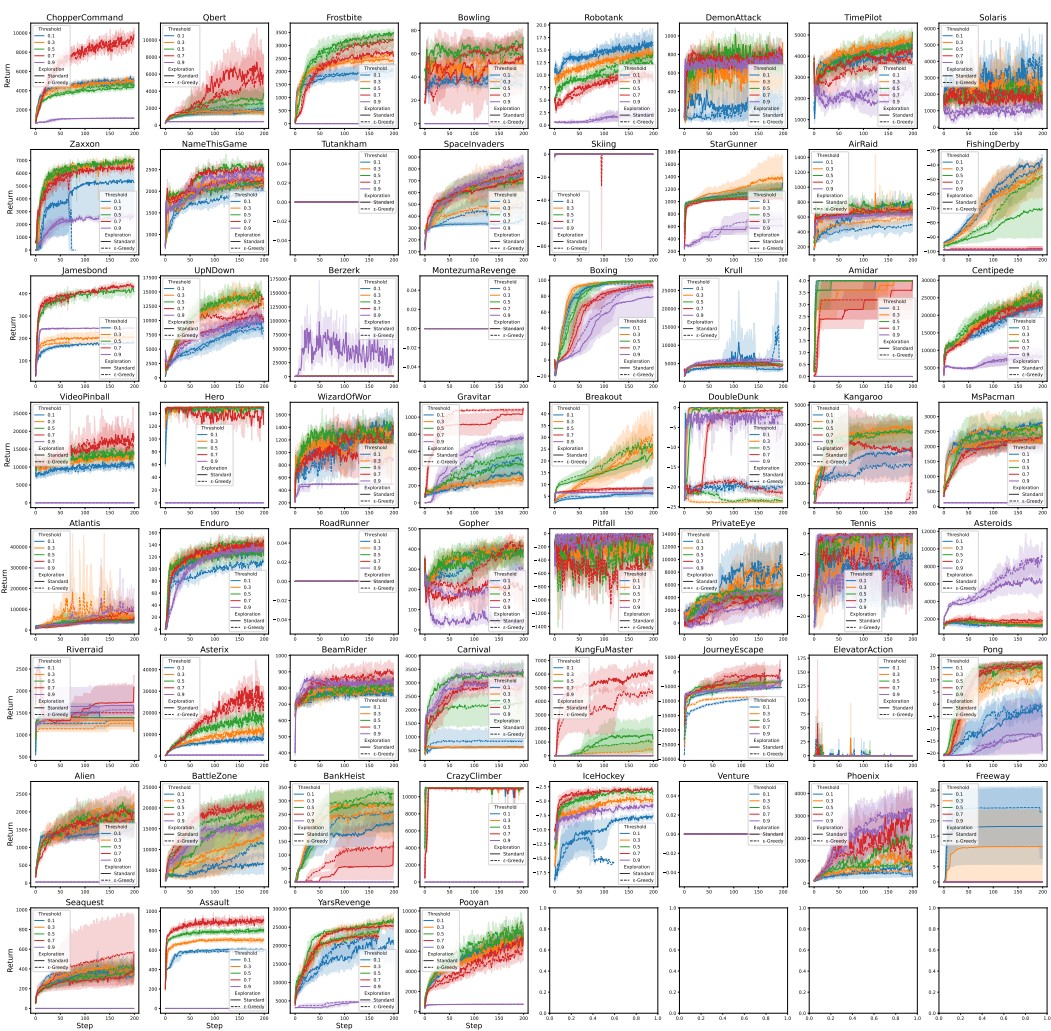

Figure 9: Per-game learning curves for agents trained on 200M.

# F    SAC-D extra results

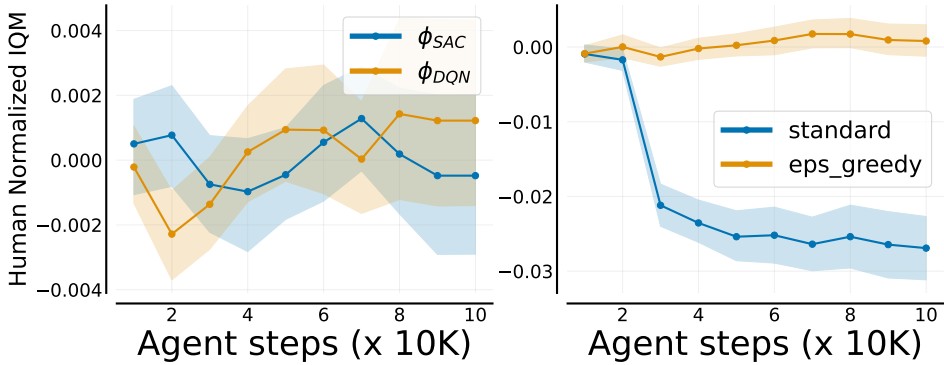

Figure 10: **Left:** Comparison of encoders on SAC-D with $\epsilon$-greedy exploration; **Right:** Comparison of exploration strategies with the $\psi_{DQN}$ encoder. Reporting IQM averaged over the 26 Atari 100K games 5 runs with 95% stratified bootstrap intervals [Agarwal et al., 2021].

