# OpenReview forum: "CALE: Continuous Arcade Learning Environment"
_NeurIPS.cc/2024/Datasets_and_Benchmarks_Track — NeurIPS 2024 Track Datasets and Benchmarks Poster_

### Official Review · Reviewer_Bcrv · 2024-07-11
**CALE**

**Rating:** 6
**Confidence:** 4
**Correctness:** The results seem correct to me.
**Clarity:** The paper is clear and easy to follow.

**Review:**

The paper introduces an interesting new perspective on the ALE suite of environments.
I think the contribution is valuable: the authors clearly make the point of why this is needed and what usage one could do with the new suite.

**Strengths:**

The paper is well written. It is easy to follow, and the contribution is clearly stated.

I believe this is a tool that is likely to be used by the community and I would encourage researchers who propose algorithms that can be used both in continuous and discrete settings to compare the results on ALE and CALE.

**Additional Feedback:**

NA

**Documentation:**

A bit more technical information about the framework would help to understand better how this can be used.

**Opportunities For Improvement:**

As a technical paper, I would have expected a bit more information on how this can currently be executed (it is written that the authors are working on integrating it within gymnasium, which is great). It would be nice to clearly state how easy it is to swap a regular ALE environment with a CALE version, what are the requirements in terms of python environment, machine etc. A statement about runtime of these environments and possibly integration (or not!) with other tools such as envpool would also be useful.

I felt that the reliance on SAC and only SAC only was the main limitation of the paper. I would encourage the authors to add more experiments with other popular algorithms (as pointed in the paper, many people will want to see a PPO version of this). Conclusions about exploration strategies etc. could perfectly be overfit to the algorithm being used, and rather than explaining these observations in detail (which I think are not strictly related to the novelty of the method) I would have liked to see how well these environments can be trained with SOTA algorithms that are known to work in continuous and discrete spaces.

From the theoretical point of view, I would have liked to see some comments about what it means to have a continuous action space where the action has a step-wise effect on the behaviour like it is the case here. For instance, given that the continuous actions have a step-wise impact on the behaviour, how would this work with Dreamer where actions are reparametrized? Another interesting thing to discuss would be how would Gaussian noise would affect the behaviour (depending on \tau) since a low amount of noise could result in the same actions (which will not be the case for e-greedy - the same epsilon frequency will give the same amount of random actions if the actions are evenly spaced in the continuous space).

**Relation To Prior Work:**

The paper is well integrated in the literature.

**Summary And Contributions:**

The paper introduces CALE, a new set of RL environments where the ALE environments are augmented with continuous action spaces.

---

> ### Author Rebuttal · Authors · 2024-08-16
>
> We thank the reviewer for their comments, and were pleased to read they found “the contribution valuable”, “well written”, and “easy to follow”. We address your concerns below.
>
> > “clearly state how easy it is to swap a regular ALE environment with a CALE version…”
>
> We have included instructions in Appendix A in the [updated manuscript](https://github.com/psc-g/CALE/blob/master/CALE.pdf) and are working with Farama to include these in the documentation and README.
>
> > “I would encourage the authors to add more experiments with other popular algorithms”
>
> Please see our general comment to all reviewers regarding the inclusion of PPO.
>
> > “ given that the continuous actions have a step-wise impact on the behaviour, how would this work with Dreamer where actions are reparametrized?”
>
> This is a very good point. In fact, the SAC implementation we are using has a similar action reparameterization with a tanh-transformed Gaussian (see [here](https://github.com/google/dopamine/blob/master/dopamine/jax/continuous_networks.py#L64-L80)). It’s possible that this is part of what is making SAC struggle. Although a full exploration is outside the scope of this paper, we have added a discussion of this in section 6.
>
> > “Another interesting thing to discuss would be how would Gaussian noise would affect the behaviour (depending on \tau) since a low amount of noise could result in the same actions (which will not be the case for e-greedy - the same epsilon frequency will give the same amount of random actions if the actions are evenly spaced in the continuous space).”
>
> In the Gaussian action parameterization used by SAC, the mean and stddev are both learned. However, we agree it would be interesting to experiment with a fixed stddev to investigate how sensitive CALE is to noise in the action parameterization. We will run this sweep for the final version of the paper.

---

### Official Review · Reviewer_jHe1 · 2024-07-17
**A useful extension to a popular benchmark**

**Rating:** 6
**Confidence:** 5

**Review:**

Overall, I think this is a useful contribution that should be shared with the broader RL community. I'm not aware of any work that has attempted to play Atari games with continuous actions. The writing and figures are clear and well organized, but I have some concerns with the paper's presentation and methodology.

My biggest concern with this paper is that it seems to overstate the impact of the contribution. The authors state that:

>  In particular, [the ALE] is only compatible with discrete-action agents. This has led to certain classes of agents, often based on Q-learning [Watkins, 1989], being evaluated on the ALE, while others, typically based on policy gradient [Sutton et al., 1999] or actor-critic [Konda and Tsitsiklis, 1999] methods, evaluated on entirely different benchmarks, such as MuJoCo [Todorov et al., 2012] or DM-Control [Tassa et al., 2018].

Many of the most popular policy-gradient and actor-critic algorithms, including A3C [1], PPO [2], and IMPALA [3], were all introduced with experiments on the ALE. SAC is one notable actor-critic method which was not originally introduced with Atari experiments. However, it has since been extended to support discrete actions [4]. A discrete version of DDPG would be very similar to DQN. There seem to be very few algorithms which cannot be applied to discrete action spaces, or for which a discrete action space variant doesn't exist. As such, although the CALE may make certain research directions easier, it seems to enable a very narrow range of research that was not previously possible.

Similarly, I'm not convinced by the comparison to other continuous control benchmarks. Although the CALE can be controlled with continuous values, it does not have the same continuous dynamics as typical physics environments, where small changes to the network's outputs generally lead to different behavior in the environment. For the CALE, changes to the agent's predictions will have no affect on the environment unless that change crosses the activation threshold. The work also suggests that the ALE has generally sparser rewards than the DeepMind Control Suite. This may be true, but the results in Figure 8 compare all 57 Atari games 3 DMC environments with 4 variants each, which happen to have dense reward structures. The figure does not include sparse reward DMC tasks such as swingup_sparse and balance_sparse, which are available in the ExoRL dataset that was used for these experiments.

Overall, I feel that the contributions of this paper are valuable to the community, but not quite to the extent that the paper suggests.

[1] Mnih, Volodymyr, et al. "Asynchronous methods for deep reinforcement learning." International conference on machine learning. PMLR, 2016.
[2] Schulman, John, et al. "Proximal policy optimization algorithms." arXiv preprint arXiv:1707.06347 (2017).
[3] Espeholt, Lasse, et al. "Impala: Scalable distributed deep-rl with importance weighted actor-learner architectures." International conference on machine learning. PMLR, 2018.
[4] Christodoulou, Petros. "Soft actor-critic for discrete action settings." arXiv preprint arXiv:1910.07207 (2019).

**Strengths:**

This work is a useful extension to a popular benchmark. It should have an immediate impact on the community due to the ubiquity of the ALE in RL research. The RL community has built a significant amount of infrastructure to support research on the ALE so I can see the CALE having a very swift, positive impact on the field. While the extension is fairly small, and could also be implemented as an environment wrapper or an additional layer to action predictions, the C implementation presented here provides the best interface for researchers.
Action space design is largely done heuristically, so the experiments comparing SAC and DQN on with different action encodings may serve as a useful reference to justify prioritizing discrete action spaces, and to motivate research into methods that perform better with continuous actions. This work also includes many interesting experiments and analyses of the ALE.

**Additional Feedback:**

I understand why the authors included section 2 to justify the continued importance of the ALE, and I found it very interesting, but I also thought it was a bit of a tangent from the purpose of the paper. It could be moved to an appendix if more space is needed for experiments.

**Clarity:**

Overall the paper is very well written. A few writing/formatting notes:
* On line 217 I think you use "Mujoco" to refer to the DMC
* Appendix D is too small to read when printed
* It's usually possible to infer which benchmark is being used in each figure, but it would be helpful to explicitly say whether the ALE or CALE was used.
* The caption for figure 6 does not mention SAC-D or the 100k vs 200M distinction which is a bit confusing.

**Correctness:**

I explain my concerns in the overall review but to summarize here:
* Since the CALE discretizes actions with a threshold, its dynamics are not truly continuous with respect to the action
* The motivation seems to draw a distinction between continuous action and discrete action methods that is rarely relevant in practice.
* The reward distribution experiments do not include some sparse reward DMC environments
* As the authors point out, they compare untuned SAC vs tuned DER and DQN, which may explain its poor performance.

**Documentation:**

The code is open-sourced and available, though it seems to be missing any unique documentation. The README is forked from the ALE and should probably be updated.

**Ethics:**

I do not suspect any ethical concerns.

**Limitations:**

The paper includes a limitations section which discusses the main limitations of the work.

**Opportunities For Improvement:**

I think that PPO experiments would make a valuable addition to the experiments in this paper. If possible, hyperparameter tuning for PPO and SAC would help to kickstart research using the CALE.

**Relation To Prior Work:**

Overall the authors thoroughly explain their work in the context of prior research. I would suggest that they mention [5], which showed that discretizing the action space for DMC environments can dramatically improve performance.

[5] Tang, Yunhao, and Shipra Agrawal. "Discretizing continuous action space for on-policy optimization." Proceedings of the aaai conference on artificial intelligence. Vol. 34. No. 04. 2020.

**Summary And Contributions:**

This work introduces CALE, a continuous action extension to the Arcade Learning Environment. It explains the history and continued importance of the ALE in RL research, and compares the CALE to other notable continuous control environments. The work also includes experiments demonstrating the impact of the discretization threshold, choice of algorithm, and exploration strategy. The authors compare the action distribution of agents trained on the ALE and CALE, and find that most actions are triggered at least some of the time in the CALE but no the ALE. Finally, they compare the reward distribution of the ALE and the DeepMind Control Suite to show that the ALE has much sparser rewards. The work concludes with a discussion of research directions enabled by the CALE.

---

> ### Author Rebuttal · Authors · 2024-08-16
>
> We thank the reviewer for their feedback on our work, and were pleased to read they found it “a useful contribution” that should have “a very swift, positive impact on the field”. We respond to the reviewers comments and concerns below.
>
> > “this paper ... seems to overstate the impact of the contribution”
>
> This is a fair point and we have modified our wording so as to not give the impression that policy-gradient and actor-critic algorithms cannot handle discrete action spaces. However, we still feel that the ability to train with both discrete and continuous actions under the same emulator opens up many exciting research opportunities, as discussed in Section 6.
>
> > “dynamics are not truly continuous with respect to the action”
>
> We will be adding support for paddles (which are truly continuous) soon, but decided to maintain consistency with the original ALE in this initial version (as discussed in the Limitations section). Further, there are also other controller types (which are also fully continuous) which the CALE would enable; these controllers, however, are not currently supported in the ALE.
>
> Having said this, however, we argue that the “non-continuity” in the current version is actually a strength of the CALE, as it provides a _different_ form of action->environment mapping that can be useful for better understanding and comparing continuous agents.
>
> > “PPO experiments would make a valuable addition to the experiments in this paper.”
>
> Please see our general comment to all reviewers regarding the implementation of PPO.
>
> > “The motivation seems to draw a distinction between continuous action and discrete action methods that is rarely relevant in practice.”
>
> While we can reword our motivation to avoid giving this impression, we do feel the unified emulation under discrete/continuous action spaces enabled by the CALE opens up lots of new research opportunities, as discussed in section 6.
>
> > “The reward distribution experiments do not include some sparse reward DMC environments”
>
> Thank you for pointing this out, we have regenerated this plot with the mentioned environments and added it to the [updated manuscript](https://github.com/psc-g/CALE/blob/master/CALE.pdf).
>
> > “I would suggest that they mention [5]”
>
> Thank you for pointing us to this relevant work, we have added this paper to our discussion.
>
> > “The README is forked from the ALE and should probably be updated.”
>
> The forked version has already been integrated into the [main branch of the ALE](https://github.com/Farama-Foundation/Arcade-Learning-Environment). We are still working with Farama to update their documentation, as well as the README.
>
> Thank you for pointing out the writing/formatting issues, we have addressed all of them in the [updated manuscript](https://github.com/psc-g/CALE/blob/master/CALE.pdf).

---

> > ### Comment · Reviewer_jHe1 · 2024-08-22
> >
> > Thanks you for taking the time to respond and update the manuscript. I would have liked to see more significant changes to the background because I feel like the CALE's place among existing benchmarks is very nuanced. For instance, the updated reward distribution plots don't seem to indicate a large difference between the ALE and DMC, but you make an interesting point that the discrete dynamics and continuous actions may pose a different challenge from existing continuous control benchmarks. I'm not sure how often step-wise changes in behavior occur in other continuous control domains, but presumably it is less common than in the CALE. I'm confident that this work will have some positive effect, being a convenient addition to the most popular RL benchmark, but I'm unsure what the size of that impact will be.

---

> > > ### Author Response · Authors · 2024-08-22
> > >
> > > Thank you for your reponse! When you say, "I would have liked to see more significant changes to the background", are you referring to section 5? We'd be happy to expand a bit more on the discrete dynamics with continuous actions (unless you had something else in mind you wanted us to expand on), with examples for domains where this may be relevant.

---

### Official Review · Reviewer_JNAR · 2024-07-22

**Rating:** 7
**Confidence:** 4
**Correctness:** Good.
**Clarity:** This paper is well written.

**Review:**

Pros

- CALE significantly extends the ALE suite, which will be an important suite for the RL community.
- The paper is well-organized with a logical flow from introduction to conclusion, making it easy to follow. The authors also provide detailed explanations of the new continuous action space and how it interfaces with the existing ALE.
- The submission outlines several important research directions, encouraging further exploration and advancements in the field

Cons

- The paper only provides baseline results for SAC, limiting the comparison with other commonly used continuous control methods like PPO. More baselines will help the community understand the property of CALE.

**Strengths:**

This quality of the submission is satisfactory and is poised to impact and advance the reinforcement learning community.

**Additional Feedback:**

See improvements.

**Documentation:**

The paper and the code are well documented.

**Opportunities For Improvement:**

Providing some additional baselines (like PPO, and TD3) would make this work more impactful.

**Relation To Prior Work:**

The paper clearly discusses the related work.

**Summary And Contributions:**

The submission introduces the Continuous Arcade Learning Environment. This is an extension of the existing ALE. The primary contribution of CALE is the addition of support for continuous action space, which enables some new research directions. This enhancement allows the benchmarking and evaluation of continuous-control agents alongside value-based agents within the same environment. The author also provides some analysis for the SAC baselines.

---

> ### Author Rebuttal · Authors · 2024-08-16
>
> We thank the reviewer for their encouraging feedback on our work, and are glad they found the paper well-organized, and CALE itself to be an important suite for the RL community.
>
> Please see our general comment to all reviewers regarding the inclusion of PPO, as well as our [updated manuscript](https://github.com/psc-g/CALE/blob/master/CALE.pdf).

---

> > ### Comment · Reviewer_JNAR · 2024-08-22
> >
> > Thanks to the author for the response. I have read the updated manuscript and am happy with it. A minor issue: the colors corresponding to the DER and SAC in Figure 6 are easily confused, replacing them with contrasting colors would improve this.

---

> > > ### Author Response · Authors · 2024-08-22
> > >
> > > Thank you for pointing out the issue with colors! We have just updated them in the [updated manuscript](https://github.com/psc-g/CALE/blob/master/CALE.pdf).

---

### Author Rebuttal · Authors · 2024-08-16

All reviewers suggested adding training runs for PPO, which we agree is a great idea. In order to ensure fair comparisons across the methods, we felt PPO should be run under the same infrastructure (Dopamine) where the other agents were run; unfortunately PPO is not included with Dopamine.

However, given your feedback we have implemented PPO in Dopamine and are evaluating it on all the experiments included in this paper, so as to have comparisons with both SAC and PPO in the final version of this submission. As stated in [1], there are _many_ details necessary to obtain a good performance with PPO; we are currently going through this blog post to ensure our Dopamine implementation includes them all. In the meantime, we have included learning curves for the current state of our PPO on the 100k benchmark in Figure 6 in the [updated manuscript](https://github.com/psc-g/CALE/blob/master/CALE.pdf); we will update this figure as we continue to integrate recommended practices from [1] to improve its scores.

Additionally, we are also running CleanRL’s implementation of PPO on the CALE, and will add both results to the final version.

Finally, we have also uploaded a revised version of our manuscript addressing all the points raised, with the changes highlighted in blue, available [here](https://github.com/psc-g/CALE/blob/master/CALE.pdf).

[1] [Shengyi Huang, "The 37 Implementation Details of Proximal Policy Optimization", ICLR Blog Track, 2022.](https://ppo-details.cleanrl.dev/2021/11/05/ppo-implementation-details/)

---

### Decision · Program_Chairs · 2024-09-26

**Decision:**

Accept (Poster)

**Comment:**

The paper introduces the Continuous Arcade Learning Environment, an extension of ALE through the addition of support for continuous action space. The extension enables benchmarking of continuous-control agents and value-based agents within the same environment.
The reviewers see an immediate impact potential on the further development of the reinforcement learning research thanks to the introduced extension. They appreciate the quality of overall quality of the paper presentation too. The authors have being running additional experiments and improved presentation aspects upon the requests of reviewers.